# Impact of the Pressure-Free Yutori Education Program on Myopia in Japan

**DOI:** 10.3390/jcm10184229

**Published:** 2021-09-17

**Authors:** Satoshi Ishiko, Hiroyuki Kagokawa, Noriko Nishikawa, Youngseok Song, Kazuhiro Sugawara, Hiroaki Nakagawa, Yuichiro Kawamura, Akitoshi Yoshida

**Affiliations:** 1Department of Medicine and Engineering Combined Research Institute, Asahikawa Medical University, Asahikawa 078-8510, Hokkaido, Japan; 2Asahikawa Red Cross Hospital, Asahikawa 070-8530, Hokkaido, Japan; hkago@asahikawa-rch.gr.jp; 3Department of Ophthalmology, Asahikawa Medical University, Asahikawa 078-8510, Hokkaido, Japan; nnori@asahikawa-med.ac.jp (N.N.); ysong@asahikawa-med.ac.jp (Y.S.); k-sugawara@asahikawa-med.ac.jp (K.S.); nakagawa@asahikawa-med.ac.jp (H.N.); pyoshida@asahikawa-med.ac.jp (A.Y.); 4Health Administration Center, Asahikawa Medical University, Asahikawa 078-8510, Hokkaido, Japan; yk5610@asahikawa-med.ac.jp

**Keywords:** prevalence of myopia, degree of myopia, high-pressure education, pressure-free education, Yutori education

## Abstract

This study aimed to investigate the influence of educational pressure on myopia. A less-intense school curriculum was introduced nationally in Japan beginning in 2012 based on a pressure-free education policy. In this retrospective observational study, a total of 1025 Japanese medical students of Asahikawa Medical University underwent measurements of the cycloplegic refractive error and axial length (AL), from 2011 to 2020. The spherical equivalent (SE) and AL were correlated significantly with the fiscal year of births (*p* = 0.004 and *p* = 0.034, respectively) only during enforcement of the system of high-pressure education. The SE and AL regression rates during the two educational approaches differed significantly (*p* = 0.004 and *p* = 0.037, respectively). The prevalence of high myopia was correlated significantly (*p* < 0.001) only during the system of high-pressure education. The regression of the prevalence rate of high myopia during the two education approaches differed significantly (*p* = 0.010). The progression rates of myopia and increased prevalence of high myopia were observed only during high-pressure education, suggesting that not only ophthalmologists but also educators and the government should work on together to control the progression of myopia.

## 1. Introduction

The prevalence rates of myopia and high myopia have increased dramatically in the past 50 to 60 years, especially in developed countries in east and southeast Asia [1,2,3,4,5]. Recently, the coronavirus disease (COVID-19) had led to an unprecedented global pandemic. To contain COVID-19, strict containment measures were imposed internationally, including social-distancing regulations, limited outdoor gatherings, school closures, and switching from in-person education to online, home-based learning. With the implementation of these measures, citizens spent more time using digital devices for entertainment and education. The rapid increase in digital screen time may potentially lead to a rise of myopia rates worldwide, especially in Asia. A meta-analysis suggested that myopia and high myopia would develop in, respectively, 50% and 10% of the world’s population by 2050 [6]. High myopia increases the risk of ocular conditions with serious visual impairment, such as retinal detachments, macular holes, glaucoma, and myopia macular degeneration [7,8,9]. In addition, early onset of myopia is associated with higher final myopia [10,11,12]. Therefore, prevention of myopia progression needs to begin at younger ages. Although the etiology of the onset and progression of myopia remains unclarified, education as one of the environmental factors has been reported to be correlated with them [13,14,15], and school curriculum consisting of greater amounts of near work is associated with a higher rate of myopia [15,16,17].

In Japan, graduating from a good university is a guarantee of joining a good company, and this has driven the “education-background society” and “exam hell”. The university entrance examinations have been the dominant factor in Japanese education. Therefore, the educational system in Japan had been characterized by cramming, e.g., rote learning, drilling, testing, etc., the so-called high-pressure educational practices. However, these have resulted in school dropout, bullying, school absenteeism, violence, and classroom collapse. To solve these educational problems, a less-intense school curriculum for the first nine years of compulsory education was introduced nationally based on a pressure-free or relaxed education policy, the so-called “Yutori” educational policy in Japan, from fiscal year (FY) 2002 [18,19]. This approach sought to create a relaxed learning environment for children by reducing classroom hours and learning content. The classroom hours gradually decreased from 8935 h to 8307 h during elementary school and junior high school until FY 2012.

Furthermore, Japan also adopted a five-day school week, with cessation of Saturday classes. Unexpectedly, this educational reform provided an opportunity to conduct a nationwide social experiment in Japan to study the impact of education on myopia.

To analyze the association between educational pressure and myopia, we investigated the refractive error, axial length (AL), and the prevalence rates of myopia and high myopia before and after the introduction of the Yutori educational policy in Japan.

## 2. Materials and Methods

### 2.1. Study Population

A total of 1025 Japanese medical students of Asahikawa Medical University participated. The students underwent ophthalmic examinations to measure the refractive error and AL and determine the status of the fundus during the clinical clerkship for ophthalmology, conducted over 10 years from April 2011 to February 2020. All investigations in this study adhered to the tenets of the Declaration of Helsinki; Institutional Review Board/Ethics Committee of Asahikawa Medical University approval was obtained. All participants provided informed consent before the examination. 

In Japan, the FY runs from 2 April to 1 April of the following year. The FYs of the student births ranged from 1961 to 1997. The Yutori educational approach started from FY 2002, which corresponded to the FY of birth 1987. Additionally, the new educational system introduced from FY 2012, which corresponded to FY of birth 1997. Therefore, we excluded nine students who were born in or later than FY 1997.

Students who had undergone laser in-situ keratomileusis (*n* = 9), an eye surgery (*n* = 2), or had a history of wearing orthokeratology contact lens (*n* = 1) were excluded. We also excluded students if there were only one or two born in a particular FY (*n* = 9). After excluding these students, 995 (97.1%) were included in this study (Figure 1).

### 2.2. Measures

Sixty minutes after instillation of 0.5% tropicamide and 0.5% phenylephrine hydrochloride in the left eyes, students underwent the refractive error measurement using an autorefractometer (TONOREF RKT-7700, Nidek, Japan) with cycloplegic refraction and the axial length (AL) measurement using a partial coherence interferometry (IOL MASTER 500, Carl Zeiss Meditec, Oberkochen, Germany). The accuracy of the measurement for the autorefractometer was set at 0.01 diopter (D). For autorefraction measurements, the results were converted to the spherical equivalent (SE) (half the amount of cylinder plus the spherical component). Myopia was defined as a SE refractive error of −0.5 D or lower. High myopia was defined as a SE refractive error of −6.0 D or lower. All examinations were performed using the same device throughout the 10-year study period.

### 2.3. Statistical Methods

Gender differences in age, SE, and AL were evaluated using the unpaired *t*-test, and differences in the prevalence rates were compared using the chi-square test. We used regression analysis and analysis of covariance to compare the regression by FY of birth between before 1987 and after 1987. Statistical analysis was performed using SPSS software, version 24.0 (IBM Corp., Armonk, New York, NY, USA). *p*-Values less than 0.05 were statistically significant. 

## 3. Results

A total of 995 students (317 women; 678 men; mean ± standard deviation, 24.8 ± 3.8 years) were included; the FYs of birth ranged from 1967 to 1996. The women were significantly (*p* = 0.016) younger than the men. The mean spherical equivalent (SE) and the mean AL, respectively, were −4.3 D and 25.59 mm (95% confidence interval (CI); −4.49 D to −4.12 D and 25.51 mm to 25.68 mm, respectively). The mean SE values in women and men did not differ significantly, while the mean AL in women was significantly (*p* < 0.001) shorter than in men. There were no significant gender differences in the prevalence rates of myopia (*p* = 0.635) and those of high myopia (*p* = 0.800) (Table 1). The SE was correlated significantly (r = 0.829, *p* < 0.001) with the AL (Figure 2).

When we divided the FYs of birth into two groups, i.e., before 1987 and 1987 and after, the SEs were correlated significantly with the FYs of birth (r = −0.213, *p* = 0.004) and decreased about 0.16 D annually during the FYs of birth before 1987. However, the SEs were not correlated significantly with the FYs of birth (*p* = 0.441) from 1987 and after. The SE regression rates based on the FYs of birth before 1987 and after differed significantly (*p* = 0.004) (Figure 3).

The ALs were correlated significantly with the FYs of birth (r = 0.157, *p* = 0.034) and increased about 0.05 mm annually during the FYs of birth before 1987. However, the ALs were not correlated significantly with the FYs of birth (*p* = 0.599) from 1987 and after. The AL regression rates based on the FYs of birth before 1987 and after differed significantly (*p* = 0.037) (Figure 4).

The myopia prevalence rates were not correlated significantly with the FYs of birth before and after 1987 (*p* = 0.428, *p* = 0.080, respectively). In contrast, the prevalence rates of high myopia were correlated significantly with the FYs of birth (r = 0.851, *p* < 0.001) and increased about 2.4% annually during the FYs of birth before 1987. However, the high myopia prevalence rate was not correlated significantly with the FYs of birth (*p* = 0.692) from 1987 and after. The regression rates of the prevalence of high myopia by FY of birth before 1987 and after differed significantly (*p* = 0.010) (Figure 5). The mean prevalence rates of myopia and high myopia during FYs of birth before 1987 were 82.5% and 18.1% (95% CI; 73.3% to 91.7% and 10.3% to 25.9%, respectively), and those of 1987 and after were 90.2% and 28.6% (95% CI; 88.2% to 92.2% and 24.1% to 33.1%, respectively).

## 4. Discussion

In the current study, we investigated the degree of myopia and the prevalence rates of myopia and high myopia before and after the introduction of the Yutori educational system in Japan. During high-pressure education, the degree of myopia and the prevalence of high myopia increased with the passage of the FY of birth; however, no myopia progression was observed after implementation of Yutori education. Because the pressure-free education policy reduced myopia progression, it would be interesting to understand the etiology of myopia.

A significant correlation was observed between the SE with AL. During high-pressure education, the SE decreased about 0.16 D annually, and the AL increased about 0.05 mm annually, which corresponded to about 3.0 D/mm. Therefore, the refractive change toward myopia was accompanied by AL elongation. In longitudinal studies among university students and medical students, the mean refractive changes were −0.11 D and −0.17 D annually [20,21]. The refractive difference followed by the birth year in the current study was similar to the individual refractive changes annually in those studies. Therefore, the difference in the refractive component among the different birth years should receive attention when myopia is investigated in cross-sectional studies.

In the current study, the prevalence of myopia was 90.2% after the introduction of Yutori education, which was independent of the FY of birth. High prevalence rates of myopia, i.e., about 90% or more, had been reported among medical or university students in Asia [4,21,22], which would result from a ceiling effect. In contrast, the prevalence of high myopia increased up to about 28.6% with the FY of birth during high-pressure education but stabilized after the introduction of Yutori education. This might have resulted from reduced educational pressure, but it also might have been the result of a ceiling effect, as the prevalence had been sufficiently high and was similar to 28.7% in medical students in Singapore [22].

A higher level of education has been associated with more myopia [14,15,23]. However, the level of education in the current study was the same because the participants were the same medical students in the same university. The time spent engaged in educational activities and the intensity of education also are important factors in myopia [24,25]. In Yutori education, the total number of classroom hours during the first nine years of compulsory education, starting from the age of seven years, was reduced gradually from 8935 h (until FY 1986) to 8307 h (in FY 1995). However, according to the survey on time use and leisure activities conducted by the Ministry of Internal Affairs and Communications in Japan (https://www.e-stat.go.jp/stat-search/files?page=1&toukei=00200533&result_page=1, accessed on 10 May 2020), the daily average time spent on schoolwork including classroom hours during elementary school and junior high school were almost the same or slightly increased from 281 and 326 min in FYs 2001 to 281 and 335 min in FY 2006, respectively. This suggests that the time after school increased, which might have been caused by the increased time spent doing homework and “cramming” or in private tutorial classes. Therefore, the length of time for education itself would be unrelated to the different tendency of myopia progression observed in this study. Further, both the study time and learning content were reduced during Yutori education. The time spent on the primary subjects, i.e., mathematics, science, social study, and Japanese language, were reduced, and integrated learning lessons, i.e., problem-solving or experiential learning to develop the ability to think and learn independently, were introduced. This reduced the time needed for cramming or rote learning and near work, such as reading and writing, even with the same classroom hours based on the curriculum. More time spent on near-work activities had been reported to increase the risk of myopia [24,25]. Therefore, the intensity of education and near work would be related to myopia progression during high-pressure education. In addition, Saturday became a holiday in the Yutori system, and students had more chances to play outdoors during the day. The effect of increased time spent outdoors on myopia prevention and slowing myopia progression had been reported [26,27,28,29]. Introduction of the Yutori education system might be related to prevention of myopia progression.

In previous birth cohort studies [2,30], an increase was reported in the prevalence of myopia in the population with a more recent birth year. The increase was widely considered to be driven by environmental factors, such as decreased time outdoors and increased near-work activities, among other factors. As societies develop, there have been systematic increases in education, but there have been parallel changes in the number of other parameters, such as living environments, including changes in population density, style of housing, pollution, diet, and lifestyle [1,6]. Recently, use of computers, smart phones, and tablets are suggested to play a role [2]. Our data demonstrated that the progression of myopia was similar to a previous report [2,6], but that occurred only during high-pressure education. Therefore, these data indicated that educational pressure is related to myopia progression. Although other environmental factors that continuously affect myopia progression would increase the myopia progression year by year, myopia did not progress after the high-pressure education was stopped. Because the Yutori education system started from FY 2002 and gradually reduced the classroom hours to FY 2010, the degree of high-pressure education would decrease year by year. Other environmental factors and the degree of high-pressure education might have been counterbalanced and resulted in stabilization of the degree of myopia after the introduction of Yutori education. Our study focused on adults; therefore, if we focused on children, who are at greater risk of developing myopia, the result might have been different. Additionally, our data showed the influence of the environmental factor in childhood. Such kinds of observational studies not only for children but also for adult should be done to help monitor the prevalence of myopia. The COVID-19 pandemic has changed educational institutions across the world, compelling to adopt and use the available technologies to enable remote learning for students. This recent rapidly increased screen time may potentially accelerate the high myopia-prevalence rate in Asia and worldwide [31]. It would be important to study the prevalence of myopia in children during the COVID-19 pandemic, but it would also be important to conduct a follow-up study on same population as they grow up.

During the compulsory education period, children would not have a choice to select their educational environment, as the school curriculum has been determined by the government. As the educational environment can influence the myopia progression, the government should appoint an ophthalmologist to the committee for the educational system assessment. However, the environment for daily life cannot be fully controlled. To control the progression of myopia, it would be important to inform parents and educators about the harmful effect of prolonged hours of near work and the beneficial effect of outdoor activities.

The Organization of Economic Cooperation and Development (OECD) Programme for International Student Assessment (PISA) measures 15-year-olds’ ability to use their reading, mathematics, and science knowledge and skills to meet real-life challenges (https://www.oecd.org/pisa/, accessed on 6 May 2020). The OECD PISA was first performed in 2000 and repeated every three years. Both countries with a high prevalence rate of myopia and those with lower prevalence rates of myopia had significant international rankings of educational performance, suggesting that high educational outcomes in PISA are not necessarily associated with an epidemic of myopia [1,2]. However, the OECD PISA results in Japanese students in 2003 and 2007 during Yutori education fell dramatically compared with those in 2000 during high-pressure education. Japan dropped from first to sixth and then to tenth place (in 2003 and 2006, respectively) in mathematical literacy, from eight to fourteenth and then fifteenth place in reading literacy, and from second to second and sixth place in science literacy. There were few racial differences, and the same language is spoken in Japan; only the educational system differed among these periods. The Japanese nation was shocked by this turndown of students’ academic abilities resulting from Yutori education. Therefore, pressure-free education might contribute to both preventing myopia progression and decreasing the educational standing. To solve this problem, a new school curriculum was introduced that abandoned the pressure-free education policy from FY 2012 after a one-year transition period in Japan. Further study is needed to clarify the effect of educational pressure on myopia progression after cessation of pressure-free education. 

The current study had some limitations. First, the current study was not completely reflective of the population in Japan because medical students who were at a higher education level were evaluated. However, we excluded the possibility of the effect of the educational level on myopia progression, as all were medical students in the same university. Second, the study population had twice the number of men compared with women in the current study. Therefore, the ratio of men to women differed between the participants in the current study and the Japanese population. Because no gender difference was observed regarding the SE, the prevalence rates of myopia and high myopia would not be affected by gender differences. Third, we studied only left eyes, although many previous studies used data from right eyes. However, no refractive difference between the left and right eyes has been reported previously. Therefore, this likely did not affect the current results.

## 5. Conclusions

In conclusion, we investigated the degree and prevalence of myopia before and after the introduction of free-pressure education in Japan. This study indicated that the program policy for compulsory education appeared to be associated with the progression of myopia. It is important to prevent myopia progression without compromising the education level. In order to control the progression of myopia, not only ophthalmologists but also educators and the government should work on together.

## Figures and Tables

**Figure 1 jcm-10-04229-f001:**
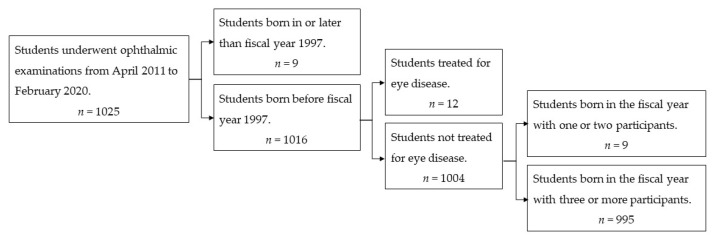
Study flowchart.

**Figure 2 jcm-10-04229-f002:**
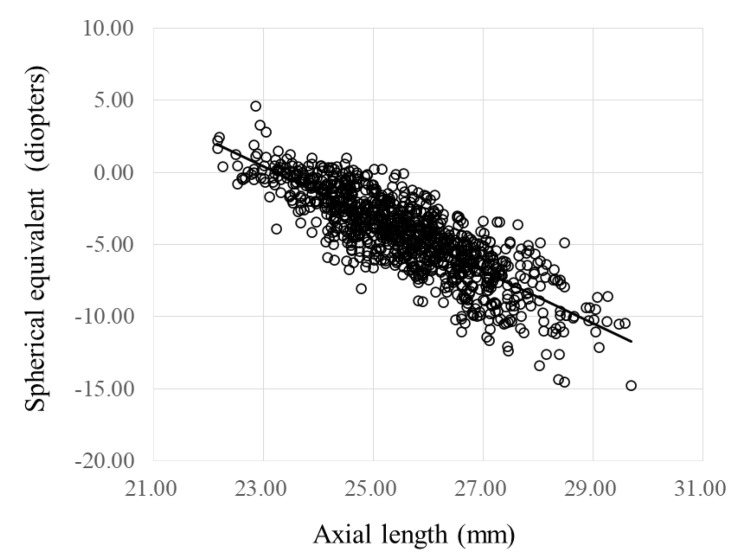
Spherical equivalent and axial length. y = 42.07 − 1.81x, r^2^ = 0.687, *p* < 0.001. ○, data of eyes in each student.

**Figure 3 jcm-10-04229-f003:**
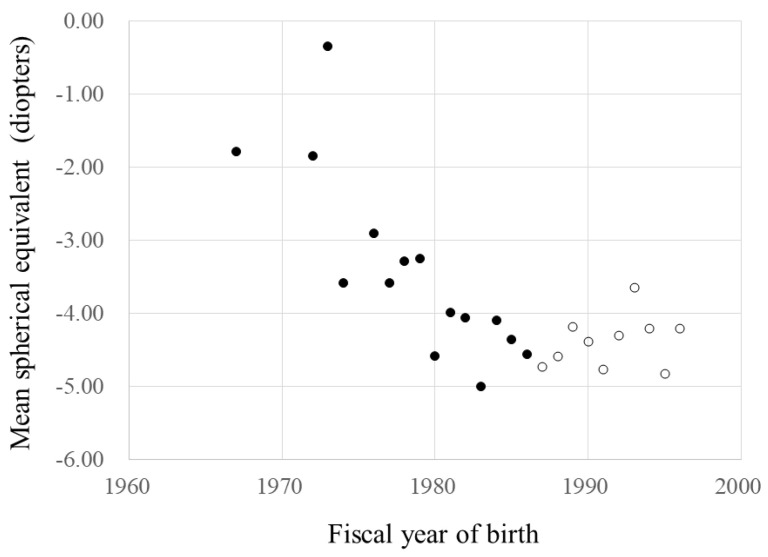
The fiscal year (FY) of birth and mean spherical equivalent. ●, data during the FYs of birth before 1987. ○, data during the FYs of birth of 1987 and after. During the FYs of birth before 1987, y = 310.0 − 0.158x, r = 0.213, *p* = 0·004. During the FYs of birth of 1987 and after, *p* =0.441. The regression of spherical equivalent by FYs of birth before 1987 and after, *p* = 0.004 (analysis of covariance (ANCOVA)).

**Figure 4 jcm-10-04229-f004:**
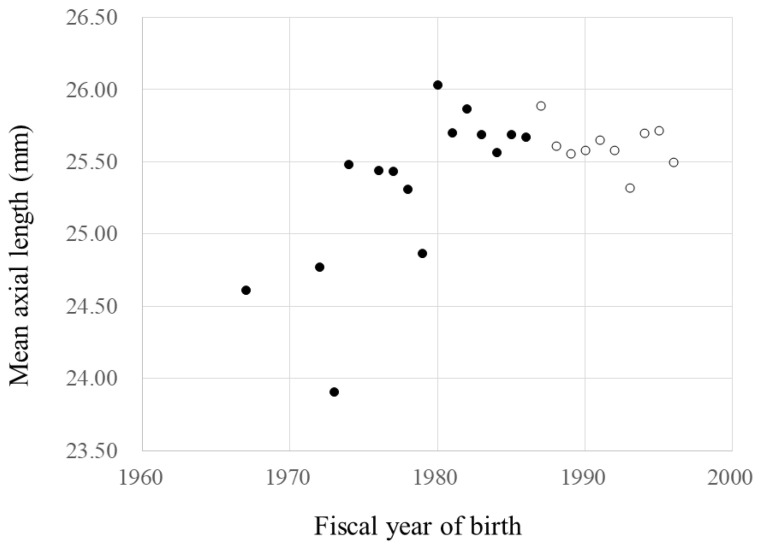
The fiscal year (FY) of birth and mean axial length. ●, data during the FYs of birth before 1987. ○, data during the FYs of birth of 1987 and after. During the FYs of birth before 1987, y = −78.9 + 0.0527x, r = 0.157, *p* = 0.034. During the FYs of birth of 1987 and after, *p* = 0.599. The regression of spherical equivalent by FYs of birth before 1987 and after, *p* = 0.004 (ANCOVA).

**Figure 5 jcm-10-04229-f005:**
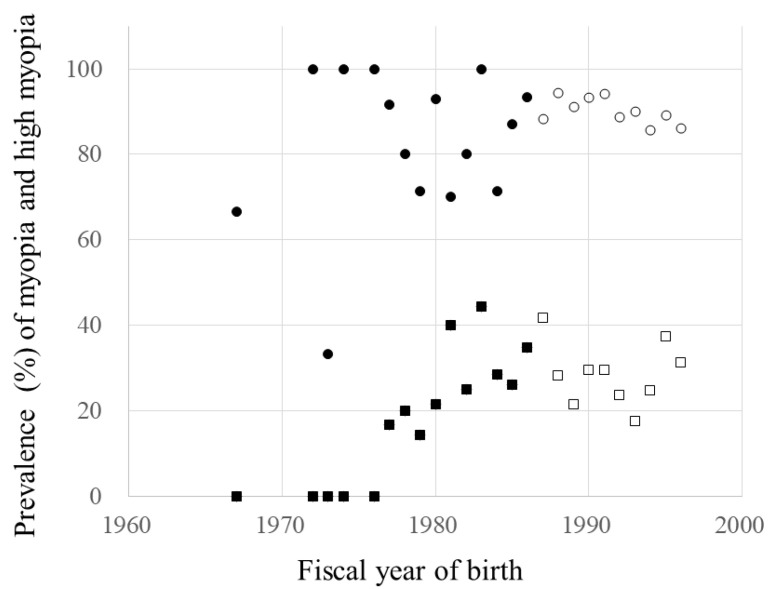
The fiscal years (FYs) of birth and prevalence of myopia and high myopia. ●, the prevalence of myopia during the FYs of birth before 1987. ○, the prevalence of myopia during the FYs of birth of 1987 and after. ■, the prevalence of high myopia during the FYs of birth before 1987. □, the prevalence of high myopia during the FYs of birth of 1987 and after. The prevalence of myopia: during the FYs of birth before 1987, *p* = 0.428; during the FYs of birth of 1987 and after, *p* = 0.080. The prevalence of high myopia: during the FYs of birth before 1987, y = −4826 + 2.45x, r = 0.851, *p* < 0.001; and during the FYs of birth of 1987 and after, *p* = 0.692. The regressions of the spherical equivalent by FYs of birth before 1987 and after, *p* = 0.010 (analysis of covariance).

**Table 1 jcm-10-04229-t001:** Prevalence rates of myopia and high myopia. * Unpaired *t*-test; ^†^ chi-square test. SE, spherical equivalent; D, diopters.

	Total (*n* = 995)	Female (*n* = 317)	Male (*n* = 678)	*p* Value
Age (years)	24.8 ± 3.8	24.4 ± 3.8	25.0 ± 3.8	0.016 *
Range	21 to 52	21 to 52	21 to 45	
SE (D)	−4.30 ± 2.96	−4.35 ± 2.94	−4.28 ± 2.97	0.733 *
Range	+4.62 to −14.73	+2.44 to −13.37	+4.62 to −14.73	
Axial length (mm)	25.59 ± 1.35	25.22 ± 1.28	25.77 ± 1.36	<0.001 *
Range	22.16 to 29.69	22.16 to 29.25	22.25 to 29.69	
Prevalence rate (%)				
Myopia (≤−0.5 D)	89.5	90.2	89.2	0.635 ^†^
High Myopia (≤−6.0 D)	27.2	27.8	27.0	0.800 ^†^

## Data Availability

The data presented in this study are available on request from the corresponding author.

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
