# Peer review of "Impact of the Pressure-Free Yutori Education Program on Myopia in Japan"

_jcm, 2021, doi:10.3390/jcm10184229_

Round 1

Reviewer 1 Report

General comments

I am very glad the authors wrote this paper.  It is an accurately, needed and useful for myopia control. This is a well written paper demonstrating the influence of educational pressure on myopia.

Introduction

The introduction would benefit from hypotheses providing a rationale.

Materials and Methods

Add a flow diagram to population and their distribution.

Results

The results section is hard to read.

Overall, I would recommend that the results be presented as a table that concisely summarises the findings of the study.

Discussion

I would like to be able to distinguish a first paragraph with a summary of what was sought and what was found in this work, a discussion of the relevant findings together with what was found in other works, an explanation of the meaning of these findings, an explanation of the implications for practice clinic and suggestions for future research.

Finally, is that there is what is the practical impact of this study?

Reviewer 2 Report

The authors investigate the influence of educational pressure on myopia. This is a relevant study, and the manuscript is well written overall.

I have some minor concerns pertaining to relevant citations as well the overall rationale of the study. The below suggested points would strengthen the background and the rationale for conducting such studies.

References 1-4 are not relevant specifically to East and Southeast Asia.

Suggest replacing with below most relevant citations to make it more specific from countries like China, India, Japan, Korea, etc.

  1. Sun J, Zhou J, Zhao P, et al. High Prevalence of Myopia and High Myopia in 5060 Chinese University Students in Shanghai. Invest Ophthalmol Vis Sci 2012; 53:7504–9.
  2. Singh NK, James RM, Yadav A, et al. Prevalence of Myopia and Associated Risk Factors in Schoolchildren in North India. Optom Vis Sci 2019;96:200–5.
  3. Jung SK, Lee JH, Kakizaki H, et al. Prevalence of Myopia and Its Association with Body Stature and Educational Level in 19-year-old Male Conscripts in Seoul, South Korea. Invest Ophthalmol Vis Sci 2012;53:5579–83.

Recently, the coronavirus disease (COVID-19) had led to an unprecedented global pandemic. To contain COVID-19, strict containment measures were imposed internationally, including social-distancing regulations, limited outdoor, school closures and switching in-person education to online home-based learning. With the implementation of these measures, citizens spent more time using digital devices for entertainment and education. This rapid increase in digital screen time may potentially lead to a rise of myopia rates worldwide, especially in Asia.

The authors should highlight this important point as well in the introduction section and how regular observational studies such as this one should be done to help monitor the prevalence. The below citations could help:

  1. Singh NK. Letter to the editor: myopia epidemic post-coronavirus disease 2019. Optom Vis Sci. 2020;97(10):911–2.

The authors must also include in their discussion how their results on adults could be generalizable? The results would be different if such a study was performed on children which is the population at greater risk of developing myopia and near work has been shown to be associated with increased myopia prevalence (Singh et al Optom Vis Sci 2019;96:200–5). This constitutes a caveat, and the authors should highlight on need for such evaluation on children.
